# Changes in Dietary Patterns through a Nutritional Intervention with a Traditional Atlantic Diet: The Galiat Randomized Controlled Trial

**DOI:** 10.3390/nu13124233

**Published:** 2021-11-25

**Authors:** Mar Calvo-Malvar, Alfonso J. Benítez-Estévez, Rosaura Leis, Juan Sánchez-Castro, Francisco Gude

**Affiliations:** 1Department of Laboratory Medicine, Complejo Hospitalario Universitario de Santiago de Compostela, 15706 Santiago de Compostela, Spain; mariadelmar.calvo.malvar@sergas.es; 2Research Methods Group, Health Research Institute of Santiago de Compostela, 15706 Santiago de Compostela, Spain; juanjose.sanchez.castro@sergas.es (J.S.-C.); francisco.gude.sampedro@sergas.es (F.G.); 3Primary Care Prevention and Health Promotion Network, Carlos III Health Institute, 28029 Madrid, Spain; 4Unit of Pediatric Gastroenterology, Hepatology and Nutrition, Pediatric Service, Complejo Hospitalario Universitario de Santiago de Compostela, 15706 Santiago de Compostela, Spain; 5Pediatric Nutrition Research Group, Health Research Institute of Santiago de Compostela, 15706 Santiago de Compostela, Spain; 6CIBEROBN, (Physiopathology of Obesity and Nutrition) Institute of Carlos III Health, 28029 Madrid, Spain; 7Unit of Investigation in Human Nutrition, Growth and Development of Galicia (GALINUT), University of Santiago de Compostela, 15782 Santiago de Compostela, Spain; 8A Estrada Primary Care Center, A Estrada, 36680 Pontevedra, Spain; 9Clinical Epidemiology and Biostatistics Unit, Complejo Hospitalario Universitario de Santiago de Compostela, 15706 Santiago de Compostela, Spain

**Keywords:** family-based randomized trial, Atlantic diet, community-focused dietary intervention, primary care, quadruple-helix, PCA, dietary patterns, body weight

## Abstract

Unhealthy dietary patterns (DPs) can lead to cardiovascular and other chronic diseases. We assessed the effects of a community-focused intervention with a traditional Atlantic diet on changes in DPs in families and the associations of these changes with weight loss. The Galiat study is a randomized, controlled trial conducted in 250 families (720 adults and children) and performed at a primary care setting with the cooperation of multiple society sectors. Over 6 months, families randomized to the intervention group received educational sessions, cooking classes, written supporting material, and foods that form part of the Atlantic diet, whereas those randomized to the control group followed their habitual lifestyle. At baseline, five DPs that explained 30.1% of variance were identified: “Caloric”, “Frieds”, “Fruits, vegetables, and dairy products”, “Alcohol”, and “Fish and boiled meals.” Compared to the controls, the intervention group showed significant improvements in “Fruits, vegetables, and dairy products” and “Fish and boiled meals” and reductions in the “Caloric” and “Frieds”. Changes in bodyweight per unit increment of “Frieds” and “Fruits, vegetables, and dairy products” scores were 0.240 kg (95% CI, 0.050–0.429) and −0.184 kg (95% CI, −0.379–0.012), respectively. We found that a culturally appropriate diet improved DPs associated with weight loss.

## 1. Introduction

Diet is one of the most important modifiable risk factors for cardiovascular disease and other chronic noncommunicable diseases, and dietary changes are a key strategy to prevent millions of deaths per year around the world [1]. However, changing dietary habits is challenging as habits are influenced by complex and intertwined societal and individual-level factors including culture, food affordability, immediate friends and family, and the surrounding community [2]. The assumption that most people would replace unhealthy dietary components in light of new scientific evidence about their associated disease risk is overly optimistic [3].

Dietary interventions that are solely based on addressing individual-level barriers such as an individual’s eating preferences without addressing the underlying sociocultural barriers may not result in sustainable weight loss or maintenance in the long term [2]. Rather, the innovative quadruple-helix model (4H) [4,5] proposes that the effective cooperation of all members of society, including public institutions, industry, academia, and citizens, is essential to promote innovative interventions where impact extends far beyond what an isolated entity can achieve on its own. In this model, the cultural aspects of the diet should be taken into account. Therefore, to ensure adequate dietary change that supports achieving the UN Sustainable Development goals [6], healthy, affordable, and sustainable diets that taste good and are culturally appropriate are recommended [7]. However, empirical evidence around the role of a 4H-model-based dietary intervention in both adequate dietary change as well as weight loss is limited.

Spain has long been associated with two well-known dietary patterns: the Mediterranean diet and the Atlantic diet, both considered healthy, affordable, and environmentally sustainable [8]. Adherence to the Atlantic diet has been recommended in many countries since the year 2000 in part due to the diet’s association with improved cardiometabolic health and lower coronary mortality [9,10,11]. However, innovative dietary interventions that promote and support adherence to the Atlantic diet are needed. Within the healthcare sector in Spain and many other nations, primary care is the initial locus that facilitates access to comprehensive healthcare and where services are staffed by multidisciplinary professionals. Services are provided to and are in collaboration with populations, and they include promotive, preventive, curative, and rehabilitative services [12]. Primary care delivers health promotion and educational interventions and fosters continuity of care and long-term relationships with health professionals, thereby increasing benefits [13,14] such as better treatment adherence [15,16].

Therefore, we hypothesized that a multilevel synergistic dietary intervention involving primary healthcare providers, families, and the community itself would help to achieve healthful eating. The aims of the present study were to investigate whether an intervention conducted at a primary care setting, based on a traditional diet, with the cooperation of all society members, can produce changes in dietary patterns and to investigate whether associations exist between changes in dietary patterns and weight loss.

## 2. Materials and Methods

### 2.1. Study Design

The Galiat study (Galicia Atlantic Diet) was a parallel-arm, community-focused dietary intervention trial designed to assess the effects of a traditional Atlantic diet on improving anthropometric variables, metabolic profile, and nutritional habits. The trial was retrospectively registered with clinicaltrials.gov (NCT02391701; https://clinicaltrials.gov/ct2/show/NCT02391701; accessed on 18 March 2015). A detailed description of the study population, participant eligibility, cluster-based randomization, sample size calculation, and dietary intervention are described in detail elsewhere [17,18]. In brief, a random representative sample (3500 male or female individuals aged 18–85 years and stratified by decades of age) was drawn from the Spanish National Health System Register of a single rural population of about 20,000 inhabitants. These individuals, called index subjects, and all relatives who shared the same home, were invited to participate. Index subject had to be living in a family unit of at least two members, male or female, and aged 3–85 years old. Exclusion criteria for the index subject were alcoholism (using the AUDIT-C questionnaire [19]), current use of lipid-lowering medication, pregnancy, major cardiovascular disease (ischemic heart disease, heart failure, peripheral vascular disease, or cerebrovascular disease), dementia, or having a predicted survival of less than 1 year. The exclusion criteria for the other members of the family were the same except undergoing lipid-lowering medication. Families (clusters) were randomly allocated (1:1) to the intervention group or to the control group. The study was approved by the Galician Autonomic Committee for Research Ethics (code 2013/531), and participants were provided written informed consent at enrollment. This manuscript conforms to CONSORT [20] reporting guidelines (Appendix A, CONSORT checklist) and the TIDieR [21] checklist (Appendix A).

### 2.2. Theoretical Model

The intervention was designed using a quadruple-helix model [4,5] and aimed to promote changes in nutritional patterns. This was achieved by a cooperation model in which citizens, businesses, research actors, and public institutions collaborated to share resources, produce knowledge, and build the confidence and flexibility required to carry the fieldwork out. The main characteristics of the study were the use of a diet congruent with the gastrocultural heritage of the study area, the Atlantic diet; the family as the intervention unit; fieldwork conducted at the primary healthcare setting; and a cooperation model with the support of the City Hall, the media, local businesses, a hostelry school, and local restaurants. The setting for participant recruitment and delivery of intervention was the primary healthcare center of a rural municipality in northwestern Spain. This setting enabled a collaborative multisector approach with existing social connections and community resources (Figure 1).

### 2.3. Dietary Intervention

The nutrition content of the intervention was based on the Atlantic diet, the traditional dietary pattern in northwestern Spain and Portugal, which is composed, above all, of home-cooked local, fresh, and minimally processed seasonal products. Although it has many elements in common with the Mediterranean diet such as high consumption of vegetables, fruits, whole grains, beans, and olive oil as key fat sources, the Atlantic diet has some differentiating characteristics such as the increased intake of fish and seafood, potatoes, bread, dried fruits like chestnuts, milk and cheese and moderate consumption of meat and wine [17,22]. The dietary intervention was aimed at modifying food behaviors in accordance with the characteristics of the Atlantic diet but not necessarily to restrict energy intake. Dietary recommendations were tailored to accommodate the preferences and nutritional needs of each participant.

During the six-month study period, families randomized to the intervention group received three educational sessions on nutrition delivered at the primary health care center by nutritionists, a cooking class given by a local chef, written supporting material including a recipe book, and food baskets (free of charge) delivered every 3 weeks with a variety of local foods characteristic of the traditional Atlantic diet (Table 1). Those randomized to the control group followed their habitual lifestyle.

### 2.4. Sociodemographic, Health, and Body Weight Data

Sociodemographic data such as age, gender, and educational level were collected from each study participant at baseline (visit 0). Health-related quality of life [23], physical activity [24], smoking habit, and education were determined with a questionnaire at the time dietary records were collected. Body weight measurements were taken in triplicate at baseline and at 6 months. Body weight to the nearest 0.1 kg was measured with a calibrated beam scale (SECA^®^ 701 model-class III, digital display, Hamburg, Germany), when subjects were standing barefoot in light clothing. Change in body weight was defined as the difference between body weight at 6 months and at baseline.

### 2.5. Dietary Assessment

Dietary intake was assessed using a 93-item food frequency questionnaire based on a Spanish adaptation [25,26] of a previously validated instrument [27,28,29]. Questionnaires were administered at baseline and at 6 months during face-to-face visits with nutritionists to assess food habits during the preceding month. For each food item, the questionnaire asked for the portion sizes. For food items which were not commonly consumed in household measures, the nutritionists showed colored photographs to help participants in describing their usual portion sizes. The responses had nine frequency categories ranging from “never or less than once per month” to “six or more times per day”. Questions about the fat content of milk products, added sugar to dairy products or beverages, fiber cereals content, and type of cooking were also integrated into the FFQ. The selected frequency category for each food item was converted to a daily intake. For example, a response of “2–4/week” was converted to 0.43 servings/day (3 servings/week). All completed questionnaires were checked by a nutritionist for accuracy and completeness.

### 2.6. Statistical Analyses

Before conducting the food pattern analysis using principal components analysis (PCA), individual food and food ingredients from the dietary records were first aggregated into groups. Where possible, foods were separated into full- and reduced-fat groups by fiber content or by cooking technique. Food groupings are specified in Appendix A.

PCA was used to reduce the 34 food groups (Appendix A) to a small number of components constituting a dietary pattern (DP) that explained the maximum fraction of the variance. Food groups were entered into the PCA as number of servings. DPs were derived by PCA of the baseline sample. After varimax orthogonal rotation, which typically associates each loading variable (food group) with one component and thereby increases interpretability, principal component scores were generated. The number of DPs retained was based on eigenvalues (0.15), the scree plot of eigenvalues, the screen test [30], and the factor interpretability. The Kaiser–Meyer–Olkin statistic was used to assess the adequacy of sampling for the predictors (food groups) used in the PCA. The Kaiser–Meyer–Olkin sampling adequacy was >0.6 for all 34 food groups (values between 0.6 and 1.0 support PCA use).

Factor loadings were calculated for each food group across the selected DP. A component score was then calculated for each participant for each of the DPs identified at both time points by multiplying the factor loadings with the corresponding standardized value for each food and summing across the 34 food items [30]. Namely, the factor score for each DP was constructed by summing observed intakes of the component food groups weighted by factor loadings, so each participant received a factor score for each identified DP at baseline and at 6 months.

To assess changes in DPs following the intervention, the scoring system derived from baseline data was applied to participants’ diets at 6 months and the dietary pattern scores were calculated for each participant [31]. The analyses were conducted in the intention-to-treat (ITT) population, including all randomized adults. Some of the FFQ items was missing in 4.5% of the participants, and body weight was missing in 4.7% of the participants. Multivariate imputation by chained equations method was used to replace missing data. A per-protocol (PP) sensitivity analysis (only adults with preintervention and postintervention data) was also performed. Baseline comparability of intervention and control arms was assessed using chi-squared tests (categorical variables) and paired Student’s *t*-tests (continuous variables). Normality of dietary patterns was investigated using distributional plots.

We used linear mixed models adjusted for age, sex, and baseline values of each DP score, with the intervention as a fixed effect and family as a random effect to estimate the impact of the intervention on DP scores. The intraclass correlation coefficient (ICC) was used to assess the correlation of measurements made on individuals of the same cluster (family). Linear mixed models adjusted for baseline body weight, age, and sex were also used to assess whether change in DPs were associated with changes in body weight at 6 months. All data were run using STATA 16. Statistical significance was set at *p* < 0.05.

## 3. Results

Of the 250 families (720 participants) that were recruited and randomized, a total of 120 families (275 adults and 71 children) in the intervention group and 111 (256 adults and 59 children) in the control group completed the trial (Figure 2). As previously reported [18], the randomization produced two groups with similar characteristics of baseline sociodemographic and potential confounding factors, including education, smoking, alcohol intake, and physical activity (*p* > 0.05) with the exception of age, which was significantly higher in the intervention group (*p* = 0.012) (Table 2).

PCA of baseline food groups identified five major DPs with the largest eigenvalues, which explained 30.3% of food intake variance and could be interpreted meaningfully in terms of nutritional characteristics (Table 3). The first dietary pattern was referred to as “Caloric” for its high loadings on high-energy drinks, processed meats, precooked food, pizza, snacks, mayonnaise and ketchup, sweets, and sunflower oil. The second “Frieds” pattern was characterized by high loadings on refined grains, fried meats, fried potatoes, fried fishes, whole-fat dairy products, sunflower oil, and sweets. The “Fruits, vegetables, and dairy products” pattern reflected high consumption of fruits, tea and herbal tea, honey, vegetables, nuts, olive oil, sweets, low-fat dairy products, and whole grains. The fourth, “Alcohol,” was characterized by high consumption of beer, liquors, wine, coffee, olive oil, and processed meats. Finally, the fifth was labeled “Fish and boiled meals” due to its high loadings on boiled meats, boiled potatoes, legumes, vegetables, boiled fishes and seafood, fried eggs, boiled eggs, sunflower oil, and fried fishes.

Dietary patterns scores were split into tertiles. Food items, expressed in servings/week by DPs, are shown in Table 4.

### 3.1. Effect of the Galiat Intervention on DP Scores

Dietary intake patterns after 6 months were analyzed by PCA using the five components extracted at baseline. Subjects in the intervention group showed significant within-group differences toward healthier scores in the five DPs. In contrast, subjects in the control group showed significant within-group differences toward healthier scores only in the “Frieds” and “Alcohol” DPs (Figure 3). Linear mixed regression analysis showed differences in favor of the intervention group in all dietary patterns with the exception of “Alcohol” (Table 5), after adjusting for baseline DPs, weight, age, and gender, with family as random effect. The sensitivity analysis using the PP population revealed similar results (Appendix A).

ICC (tendency for the measurements of members of the same family to be more similar to each other than to people from other families) was higher than 0.20 for all DPs, with the exception of “Liquors” (ICC ≤ 0.10).

### 3.2. Association between DP Changes and Body Weight

Individuals in the intervention group lost weight (−0.8 kg; 95% CI, −1.1 to −0.5) whereas individuals in the control group increased weight (0.4 kg; 95% CI, 0.1–0.7) (adjusted mean difference −1.1 kg, *p* < 0.001). The association between changes in each DP and body weight changes is shown in Table 6. Changes in “Frieds” were significantly and positively related to changes in body weight (0.240; 95% CI, 0.050–0.429). Increased intake of the “Fruits, vegetables, and dairy products” DP tended to be associated with decreased body weigh but did not reach statistical significance (*p* = 0.063). The sensitivity analysis using the PP population showed similar results, but increased intake in “Caloric” pattern was also found to be associated with weight increased. (Appendix A). Collectively, these findings suggest that the intervention resulted in significant DP changes, which in turn contributed to changes in body weight.

## 4. Discussion

In this study, we report that an innovative randomized trial with a nutritional intervention to promote and support a traditional Atlantic diet modified family nutritional behaviors with consequences on body weight. Specifically, shifting dietary intake from a highly caloric and fried-foods pattern to a plant-based pattern was associated with a decrease in body weight. The uniqueness of this trial is threefold and has implications on changing DPs in the real-world context. First, the dietary intervention focused on changing and measuring DPs as a whole instead of reducing caloric intake only or reducing selected groups of foods. Second, family members were included in the intervention as participants instead of focusing only on one person in the household. Third, the intervention itself was not limited to providing nutrition education but rather focused on developing the participants’ skills and knowledge necessary to change DPs in a personalized manner. Collectively, our results indicate the suitability of an intervention program conducted in a “real-world context” based on a combination of strategies adapted to local needs and local participants, with buy-in from family members, physicians, and local collaborators.

### 4.1. Mechanism and Explanations

It is useful to speculate on success factors of the Galiat intervention to aid the future implementation of community interventions. Our scientific strategy was based on a collaborative multisector approach with local stakeholders comprising public organizations involved in social and health services, family physicians, trained nutritionists, nurses, local businesses, support services (public and private), and families. This model helped us identify opportunities to promote changes in nutritional patterns by recognizing the individual and family (attitudes, beliefs, knowledge, and behaviors), interpersonal (relationships, social networks, cultural context), and community environmental (primary care center, City Hall, local businesses, local media) factors that may influence one’s ability to promote changes in nutritional patterns. Beyond the scientific strategy, an informal participatory research approach was followed (research-action-participation) with the aim of promoting mutual learning among researchers, health and City Hall workers, and local collaborators.

Fieldwork performed at a primary healthcare center seemed to be an appropriate setting for preventing and intervening to reduce disease risk factors. We used families as the unit of randomization and intervention since it is within the family that nutritional and lifestyle habits are learned. The values of the ICCs indicate that the cluster (family) factor had an important influence on the mean change dietary pattern scores, especially in “Fish and boiled meals” and “Frieds” dietary patterns, supporting the effectiveness of family-based approaches for implementing healthful dietary changes. We adapted the intervention to local contexts, and nutrition education sessions based on the traditional diet of the study area were an important part of the intervention. The cultural connection with the culinary traditions of the southern European Atlantic coasts was a positive motivational influence for participants to introduce healthy dietary changes. Indeed, the current consumption patterns in northwestern Spain are far from the traditional Atlantic diet, and recommendations to improve adherence to the Atlantic diet are timely [32]. This is consistent with our findings where none of the patterns obtained was completely healthy (e.g., the “Fruits, vegetables, and dairy products” pattern included sweets and the “Fish and boiled meals” pattern included sunflower oil and fried food).

### 4.2. Results in Relation with Other Studies

In relation to the eating patterns obtained, the “Fruits, vegetables, and dairy products” pattern was similar to the “Prudent” [33,34,35,36], “Healthy” [37], or “Fruit and vegetables” [38,39,40,41] patterns described in other studies and characterized by a high intake of vegetables, fruits, legumes, and fish. The “Caloric” pattern was similar to the previously described “Western” pattern [35,36,37] with high consumption of processed meat, refined cereals, pastries, and chips. Nevertheless, the “Fruits, vegetables, and dairy products” pattern obtained in our study included components characteristic of the traditional Atlantic diet such as high intake of honey, nuts, and olive oil [22]. Another difference from other diets includes the large factor load of foods, which are steamed, boiled, baked, grilled, or stewed rather than fried [17]. In our study, the intervention arm subjects showed a significant shift toward the “Fruits, vegetables, and dairy products” and “Fish and boiled meals” patterns, which corroborates the effectiveness of the nutritional and gastronomic education sessions based on the traditional Atlantic diet. This effect was not observed in the control group.

Whether or not this shift in dietary patterns results in sustainable weight loss and maintenance merits further investigation. Systematic reviews of feeding trials and prospective population studies of dietary patterns and chronic disease risk have shown that dietary patterns rich in plant foods, fish, and seafood that preferably include vegetable oils and low-fat dairy products are associated with a lower risk of most chronic diseases. By contrast, Western-type dietary patterns with food products low in essential nutrients and high in energy, like sugar-sweetened beverages, sweets, refined cereals, and solid fats (e.g., butter), and high in red and processed meats are associated with adverse health effects [42,43,44,45,46]. Our findings suggest that shifting eating patterns to lower intake of fried foods and alcohol while increasing intake of unprocessed fruits and vegetables might contribute to weight loss.

### 4.3. Strengths and Limitations

We targeted families representative of the Spanish population, selected intentionally from a community of moderate socioeconomic and educational level to strengthen the generalizability of our study. Although the randomized study design, high retention rates, and focus on dietary pattern changes as a whole were strengths of this study, our findings should be interpreted while accounting for several limitations. First, focusing on a single rural population (of about 20,000 inhabitants) and using a traditional diet based on local, fresh, and seasonal products can limit the generalizability of the study. However, the overarching finding that people’s health-related behaviors are heavily influenced by the environment around them and their cultural context is a first step in implementing effective models to support health in other populations. Second, contamination bias might be present. Since the study was performed in a small town and became the subject of local and national media interest, some individuals and families in the control group may have adopted some of the lifestyle, habits, or food patterns directed at the intervention group. Indeed, we did find small decreases in the “Caloric”, “Frieds”, and “Alcohol” dietary patterns in the control group, which might have deviated estimates of the effect of the intervention toward the null hypothesis, thereby reducing the differences between the control and intervention arms. Third, the strategy of donating food baskets was an incentive for participation and retention in the study. However, this strategy could affect the generalizability of our intervention to populations in which access to or affordability of local, fresh, and seasonal foods might be a barrier. Fourth, we employed the adapted version of the validated and reproducibility tested FFQ used in the IDEFICS study in children and adolescents [29], but, to our knowledge, this questionnaire has not been validated in an adult population. Nevertheless, this questionnaire has also been used in Spanish families who participated in a large multicountry longitudinal cohort study that included a sample of 5291 adults, 3082 adolescents, and 3834 children [47,48].

Finally, the principal component analysis involved several subjective but important decisions, including the consolidation of food items into food groups, the number of factors extracted, the method of rotation, the labeling of the components [33,49], and the decision of whether variables should be adjusted for total energy intake or not [34]. In this study, we chose to use the number of servings unadjusted for total energy because it was not clear which method would be best to handle these types of variables using pattern analysis [34]. Finally, dietary patterns may differ by sex, race, cultural group, and geographical region; thus, dietary patterns extracted of our study population may not be similar to other populations. Regardless, the methodological contribution of this work highlights how changes in dietary intake can be identified and measured holistically, providing an innovative approach that can be used in future studies aimed at assessing dietary intake as an outcome.

## 5. Conclusions

Our intervention based on a traditional diet and conducted at a primary care setting with the cooperation of multiple society sectors, including the family as an intervention unit, improved dietary patterns associated with weight loss. By fostering interactions between primary care professionals and community services, our community-based approach promises to be a powerful approach to engage the primary healthcare system and other health stakeholders to become more efficient, accountable, resilient, sustainable, and committed to the empowerment of communities, with the ultimate goal of improving the health and well-being of citizens.

## Figures and Tables

**Figure 1 nutrients-13-04233-f001:**
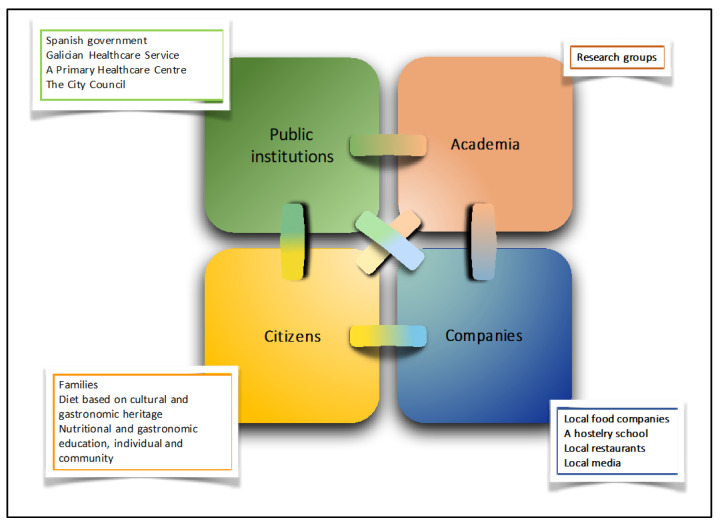
Quadruple-helix cooperation model involving four helixes: public institutions, researchers, businesses, and citizens.

**Figure 2 nutrients-13-04233-f002:**
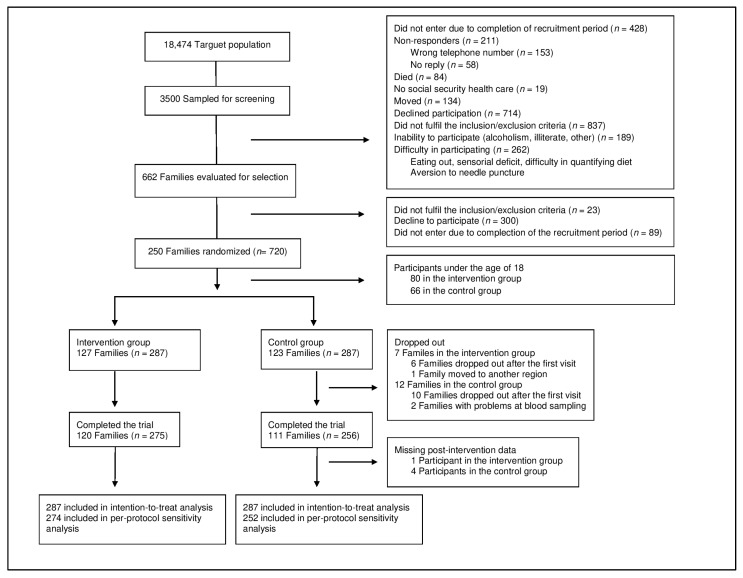
Overview of the study population.

**Figure 3 nutrients-13-04233-f003:**
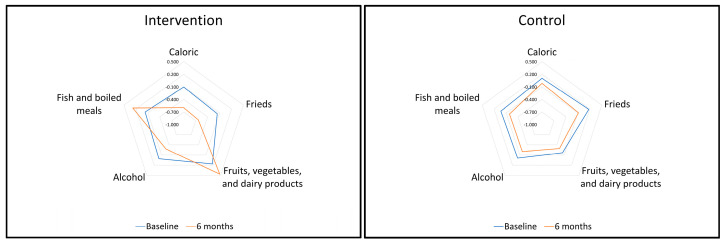
Changes in dietary pattern scores after the 6-month intervention.

**Table 1 nutrients-13-04233-t001:** Intervention components.

	Intervention Components
Basal (time 0)	Nutrition education course (30–40 min) provided to individual families by nutritionists with recommendations for adults and children: information about the Atlantic diet and the food pyramid, benefits of physical activity and eating five meals per day; information on how to prepare food menus, how and why to limit sedentary activity, and how to use the education material provided. Distribute a book with educational material and recipes. Distribute a diary with food delivery dates (every 3 weeks). Start delivery of food packages (adapted for the number of family members in one household).
Following the basal visit	Group session (2 h) by researchers and nutritionists to explain the influence of lifestyle on health, how to change to a healthier diet, importance of physical activity, characteristics of the traditional Atlantic diet, and patterns for designing a healthy diet. Cooking class led by teaching chefs; recommendations on portion size.
3 months	Nutrition education course (30–40 min) provided to individual families by nutritionists to strengthen knowledge and messages, use of educational material and recipes; to discuss realistic expectations for behavior change; to develop individual goals, behavior skills, and action plans; and to identify barriers and resolve doubts. Distribute a diary with new food delivery dates (every 3 weeks) and delivery of food packages (adapted for the number of family members).
6 months	End of the intervention. Last nutrition education course (30–40 min) provided to individual families by nutritionists to review progress and renew individual and family action plans for the future.

**Table 2 nutrients-13-04233-t002:** Baseline characteristics of the study participants (intention-to-treat data set).

	Study Sample		
Characteristic	Control Arm	Intervention Arm	*p*-Value
Families/study subjects. *N*	124/287	126/287	0.866
Participants per family. Mean ± SD	2.3 ± 0.8	2.3 ± 0.7	0.297
Male sex. *N* (%)	114 (39.7)	117 (40.8)	0.798
Age. Years. Mean ± SD	45.3 ± 15.4	48.2 ± 15.8	0.012
Marital status. *N* (%)			0.310
Married/with partner	194 (67.8)	210 (73.4)	
Divorced/separated/widowed	33 (11.5)	28 (9.8)	
Single	59 (20.6)	48 (16.8)	
Educational level. *N* (%)			0.520
None	29 (10.1)	30 (10.5)	
Elementary	120 (41.8)	103 (35.9)	
Secondary	91 (31.7)	103 (35.9)	
University or higher	47 (16.4)	51 (17.7)	
Employment status. *N* (%)			0.207
Employed	148 (51.6)	139 (48.4)	
Retired	40 (13.9)	56 (19.5)	
Other	99 (34.5)	92 (32.1)	
Smoking status. *N* (%)			0.106
Never smoker	128 (44.6)	120 (41.8)	
Ex-smoker	50 (17.4)	71 (24.7)	
Current smoker	109 (38.0)	96 (33.5)	
Alcohol intake. *N* (%)			0.789
Abstainers	123 (43.5)	124 (43.2)	
Light drinkers (1–140 g/week)	132 (46.6)	130 (45.3)	
Heavy drinkers (>140 g/week)	28 (9.9)	33 (11.5)	
Comorbidities. *N* (%)			
Cardiovascular disease	42 (16.0)	49 (18.3)	0.424
Cerebrovascular accident	3 (1.1)	3 (1.1)	1.000
Diabetes	16 (5.9)	16 (5.9)	1.000
Current medications. *N* (%)			
Cholesterol-lowering	23 (8.7)	32 (12.5)	0.202
Antihypertensives	44 (18.1)	56 (24.2)	0.187
Health-related quality of life (SF-12). *N* (%)			
Physical component summary	48.6 ± 9.3	47.3 ± 10.1	0.057
Mental component summary	51.2 ± 10.1	52.1 ± 8.8	0.120
International Physical Activity Questionnaire. *N* (%)			0.182
Inactive	56 (19.8)	44 (15.3)	
Minimally active	68 (24.6)	85 (29.6)	
Active	163 (55.6)	158 (55.1)	

No significant differences found between the two groups (chi-squared test for differences between categorical variables and Student’s *t*-test for continuous variables), with the exception of age (*p* = 0.012). SF-12, 12-item short form health survey.

**Table 3 nutrients-13-04233-t003:** Structures of the dietary patterns identified at baseline.

Dietary Patterns	Factor Loadings ^1^		Assigned Name	Variance Explained (%)
	Food Groups	Factor Loadings		
Component 1	High-energy drinks	0.37	Caloric	7.6
	Processed meats	0.35		
	Precooked food	0.35		
	Pizza	0.32		
	Salty snacks	0.31		
	Mayonnaise and ketchup	0.26		
	Sweets	0.26		
	Wine	−0.25		
Component 2	Refined grains	0.39	Frieds	6.9
	Fried meats	0.38		
	Fried potatoes	0.35		
	Fried fishes	0.28		
	Whole-fat dairy products	0.27		
	Sunflower oil	0.20		
	Sweets	0.20		
	Low-fat dairy products	−0.20		
	Whole grains	−0.37		
Component 3	Fruits	0.45	Fruits, vegetables, and dairy products	5.8
	Tea, herbal tea	0.33		
	Honey	0.32		
	Vegetables	0.31		
	Nuts	0.30		
	Olive oil	0.27		
	Sweets	0.22		
	Low-fat dairy products	0.20		
	Whole grains	0.20		
Component 4	Beer	0.51	Alcohol	5.1
	Liquors	0.51		
	Wine	0.35		
	Coffee	0.29		
	Olive oil	0.25		
	Processed meats	0.23		
Component 5	Cooked, steamed, roasted meats	0.48	Fish and boiled meals	4.9
	Boiled potatoes	0.32		
	Legumes	0.29		
	Vegetables	0.28		
	Boiled fishes and seafood	0.27		
	Fried or scrambled eggs	0.26		
	Boiled or poached eggs	0.22		
	Sunflower oil	0.21		
	Fried fishes	0.20		

^1^ Food groups based on intention-to-treat data set and ordered by size of loading coefficients. Only food groups with factor loadings ≥0.20 are presented.

**Table 4 nutrients-13-04233-t004:** Weekly intake of food items by tertiles of dietary patterns at baseline.

Dietary Pattern	T1	T2	T3
**Caloric**			
High-energy drinks	0.47	1.61	5.08
Processed meats	1.21	2.51	4.29
Precooked food	0.00	0.02	0.51
Pizza	0.01	0.05	0.55
Salty snacks	0.04	0.21	1.00
Mayonnaise and ketchup	0.10	0.38	1.82
Sweets	6.96	11.74	15.94
Wine	5.48	2.68	1.36
**Frieds**			
Refined grains	12.29	17.82	21.11
Fried meats	0.42	0.98	2.79
Fried potatoes	0.64	1.27	2.70
Fried fishes	0.59	1.30	1.84
Whole-fat dairy products	6.68	9.48	13.94
Sunflower oil	0.66	1.24	2.64
Sweets	8.63	10.80	15.26
Low-fat dairy products	11.88	8.30	6.19
Whole grains	4.98	0.81	0.43
**Fruits, vegetables, and dairy products**			
Fruits	5.00	10.91	14.33
Tea. herbal tea	0.73	1.43	4.46
Honey	0.01	0.18	1.24
Vegetables	5.19	7.24	9.92
Nuts	0.37	0.98	1.81
Olive oil	6.75	10.09	11.26
Sweets	9.06	11.57	14.07
Low-fat dairy products	6.34	8.76	11.14
Whole grains	0.91	1.65	3.58
**Alcohol**			
Beer	0.16	0.60	3.53
Liquors	0.02	0.12	0.86
Wine	0.65	2.05	6.65
Coffee	3.22	7.01	9.16
Olive oil	7.16	10.07	10.90
Processed meats	2.01	2.64	3.39
**Fish and boiled meals**			
Cooked, steamed, roasted meats	2.75	4.11	6.12
Boiled potatoes	1.59	2.03	3.10
Legumes	0.50	0.70	1.38
Vegetables	5.82	7.06	9.48
Boiled fishes and seafood	1.56	1.83	3.00
Fried or scrambled eggs	1.11	1.63	1.86
Boiled or poached eggs	0.64	1.05	1.24
Sunflower oil	0.96	1.49	2.09
Fried fishes	1.03	1.09	1.61

T, tertile. The name of the dietary patterns in bold.

**Table 5 nutrients-13-04233-t005:** Comparison of dietary pattern differences between the intervention and control groups after 6 months.

	Intervention Group		Control Group		Adjusted Mean Differences	*p*-Value	ICC
	Baseline	6 Months	Baseline	6 Months			
Caloric	−0.104 (−0.274, 0.066)	−0.597 (−0.740, 0.455)	0.106 (−0.096, 0.307)	−0.015 (−0.189, 0.159)	−0.443 (−0.667, −0.220)	<0.001	0.348
Frieds	−0.158 (−0.342, 0.025)	−0.635 (−0.788, 0.482)	0.174 (−0.342, 0.025)	−0.085 (−0.256, 0.087)	−0.376 (−0.606, −0.146)	0.001	0.389
Fruits, vegetables, and dairy products	0.165 (−0.006, 0.336)	0.462 (0.274, 0.650)	−0.168 (−0.321, 0.015)	−0.291 (−0.471, −0.111)	0.545 (0.297, 0.793)	<0.001	0.352
Alcohol	0.009 (−0.140, 0.159)	−0.278 (−0.390, 0.166)	−0.009 (−0.167, 0.147)	−0.201 (−0.348, −0.054)	−0.106 (−0.271, 0.058)	0.205	0.212
Fish and boiled meals	−0.032 (−0.181, 0.116)	0.279 (0.120. 0.438)	0.032 (−0.122, 0.185)	−0.184 (−0.345, −0.023)	0.491 (0.223, 0.759)	<0.001	0.433

Values used from the intention-to-treat data set. Values are presented as mean (95% confidence interval). Adjusted mean differences are differences between groups after 6 months for each dietary pattern which were estimated using linear mixed models adjusted by baseline values, age, and gender, with family as random effect. ICC, intraclass correlation coefficient.

**Table 6 nutrients-13-04233-t006:** Relation between baseline to 6-month changes in dietary pattern scores and body weight using the entire sample independent of group allocation (intention-to-treat data set).

Change in Food Pattern Score ^1^	Coefficient (95% CI)	*p*
Caloric	0.146 (−0.030, 0.332)	0.103
Frieds	0.240 (0.050, 0.429)	0.013
Fruits, vegetables, and dairy products	−0.184 (−0.379, 0.012)	0.063
Alcohol	0.026 (−0.185, 0.237)	0.812
Fish and boiled meals	−0.099 (−0.262, 0.064)	0.234

^1^ Food pattern exposure variables are measured as change in factor score between baseline and 6 months. All food patterns are tested together in the same model. 95% CI, 95% confidence interval.

## Data Availability

The data sets generated and analyzed during the current study will be made available through a publicly accessible repository on publication at the Runa Digital Repository (runa.sergas.gal). To gain access, data requestors will need to sign a data access agreement. Proposals should be directed to alfonsojavier.benitez.estevez@sergas.es and/or francisco.gude.sampedro@sergas.es.

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
