# Peer review of "Changes in Dietary Patterns through a Nutritional Intervention with a Traditional Atlantic Diet: The Galiat Randomized Controlled Trial"

_nutrients, 2021, doi:10.3390/nu13124233_

Round 1

Reviewer 1 Report

Based on SDGs (Sustainable Development Goals), connecting the quadruple-helix, public institutions / industry / academia / citizens, ensuring healthy, affordable, and sustainable diets, achieving healthful eating, is important public health issue for population health promotion.

Materials and Methods

Line 153-154: The authors showed the reproducibility and validity of this 93-item FFQ instrument, but those study population of cited references (#27, #28, and #29) were children population. Please explain why this questionnaire is suitable to the middle-aged persons in this family-based study.

Line 193-194: Please indicate how many variables and observations were imputated? Sensitivity analysis should perform for original and imputated data.

Results

  1. In the footnote of table 2, authors stated that "No significant differences found between the two groups". I doubts about this. For example, the age difference between groups, the calculated t-value is around 2.23 and the p-value is between 0.02-0.05. In table 2, please provide the p values for Chi-squared test and student's t test.
  2. The analysis results interpreted the impact of "DP score" on bodyweight, but the information was difficult to be translated practically into the dietary recommendation, such as the frequency and amount of dietary intake in each food item. It is recommended to add a table to show the following information:

(1) Define the "low", "middle", "high" groups, according to the DP score tertile of "Caloric" or "Fried" dietary patterns.

(2) Shows the baseline dp score, intake frequency and intake amounts (each food item in that DP) in the low / middle / high groups.

(3) Based on the above information, practical dietary recommendations can be drawn.

Reviewer 2 Report

It is a nice study highlighting and reporting the importance of community based program in promoting health benefits by encouraging adoption of traditionally healthy food habits.

The manuscript is well written, with results stating the main outcomes with appropriate statistical analysis and controls. The findings are sufficiently discussed and compared with the relevant literature.

Author Response

Dear Reviewer,

Thank you very much for your kind words.